# Microscopic parasite malaria classification using best feature selection based on generalized normal distribution optimization



Javeria Amin[1], Muhammad Almas Anjum[2], Abraz Ahmad[1], Muhammad Irfan Sharif[3], Seifedine Kadry[4,5,6,7] and Jungeun Kim[8]

[1] University of Wah, Department of Computer Science, Wah Cantt, Pakistan
[2] National University of Technology (NUTECH), Islamabad, Pakistan
[3] Department of Information Sciences, University of Education Lahore, Jauharabad Campus, Jauharabad, Pakistan
[4] Noroff University College, Kristiansand, Norway
[5] Artificial Intelligence Research Center (AIRC), Ajman University, Ajman, UAE
[6] MEU Research Unit, Middle East University, Amman, Jordan
[7] Department of Electrical and Computer Engineering, Lebanese American University, Byblos, Lebanon
[8] Department of Software, Kongju National University, Cheonan, Korea

Corresponding authors
Muhammad Irfan Sharif,
irfan.sharif@ue.edu.pk
Jungeun Kim, jekim@kongju.ac.kr

## ABSTRACT

Malaria disease can indeed be fatal if not identified and treated promptly. Due to advancements in the malaria diagnostic process, microscopy techniques are employed for blood cell analysis. Unfortunately, the diagnostic process of malaria *via* microscopy depends on microscopic skills. To overcome such issues, machine/deep learning algorithms can be proposed for more accurate and efficient detection of malaria. Therefore, a method is proposed for classifying malaria parasites that consist of three phases. The bilateral filter is applied to enhance image quality. After that shape-based and deep features are extracted. In shape-based pyramid histograms of oriented gradients (PHOG) features are derived with the dimension of $N \times 300$. Deep features are derived from the residual network (ResNet)-50, and ResNet-18 at fully connected layers having the dimension of $N \times 1,000$ respectively. The features obtained are fused serially, resulting in a dimensionality of $N \times 2,300$. From this set, $N \times 498$ features are chosen using the generalized normal distribution optimization (GNDO) method. The proposed method is accessed on a microscopic malarial parasite imaging dataset providing 99% classification accuracy which is better than as compared to recently published work.

## INTRODUCTION

Malaria is caused due to climate factors such as rainfall, humidity, and temperature. It is transmitted in subtropical and tropical regions where anopheles mosquitoes might survive and multiply. The growth cycle of the parasite malaria might be accomplished in mosquitoes (*Amin et al., 2022d*). There are numerous additional diseases with symptoms similar to malaria. As a result of the initial misdiagnoses of the disease, this may result in

late medical treatment for malaria, and the delay may cause malaria to become serious. Furthermore, malaria at a serious stage may have some complications, such as hyperpyrexia, convulsion, and hypoglycemia. According to a World Health Organization (WHO) report published in 2017, approximately 435,000 people died of malaria (*Molina et al., 2020*), and in 2018, the death rate was 405,000. Malaria affected approximately 67% of children below the age of 5 years in both years. The test of peripheral blood samples is very useful and helpful for detecting malaria. However, it is extremely difficult to manually conduct this test because it requires a high level of competence from the operator. The best approach to obtain precise information and findings from the test blood samples is the use of image processing techniques. Image processing provides useful information on cell morphology (*Loddo et al., 2018*). Pathologists conduct microscopic slide examinations (MSE), which is now the most widely used method for disease diagnosis. Moreover, the performance of MSE is dependent on the pathologists' expertise. A minor error from a pathologist can result in significant modification and inaccuracy (*Sharif et al., 2020*). Malaria spreads in two ways. The first is through female mosquito bites, whereas the second is through a blood transfusion. To stop and limit the spread of malaria, an accurate and timely diagnosis is necessary. Manual detection approaches were not suitable in rural areas where malaria has a high tendency to occur due to the lack of experience and technical skills of the examiner. Machine learning (ML) requires less training and provides minimum accuracy (*Loh et al., 2021*). That model might be trained on a smaller amount of datasets it needs more intervention from humans to learn and correct. The technique of transforming raw data into numerical values while preserving the original content of the data is known as feature extraction (*Sharif et al., 2020*). Feature extraction leads to better human interpretation (*Amin et al., 2022c*). The two main forms of feature extraction are handcrafted and deep learning. Data scientists manually extract features using the handcrafted feature extraction method, whereas deep learning features are learned automatically in the form of a pipeline (*Amin et al., 2021a*, *2022e*, *2022b*, *2021c*; *Zafar et al., 2023*). The texture and intensity features, such as Harlick texture feature, fractural, flat textural, gradient texture, Laplacian texture, color channel intensity, wavelet feature, and color autocorrelogram, are examined for malaria cell classification (*Das, Mukherjee & Chakraborty, 2015*). The textural and morphological features are extracted and applied using different classifiers that provide a maximum accuracy of 96.84% (*Das, Maiti & Chakraborty, 2015*; *Amin et al., 2021b*, *2022a*). The textural features are based on short- and long-run emphasis which achieves an accuracy of 90% (*Das, Maiti & Chakraborty, 2015*; *Špringl, 2009*). A model is suggested that detects infected erythrocytes by extracting flat texture features with an accuracy of 90% (*Arowolo et al., 2020*). Deep learning (DL) models require the maximum amount of data for training to provide better accuracy. DL models learn from their environment and previous mistakes (*Rajaraman, 2018*). You look once (YOLO)-v3 and YOLO-v4 models are employed present for classification where features are extracted using inception-v2 (*Abdurahman, Fante & Aliy, 2021*). However input malaria images have noise, poor contrast, lighting difficulties, and stains (*Poostchi et al., 2018*). Therefore, informative feature selection for malaria classification remains a

difficult task due to low-quality images, and quick variations, all of which have been addressed in this proposed work.

### Research motivation and contribution

Malaria is the most prevalent disease. This illness has affected thousands of people worldwide. People die because of the late diagnosis of this disease. Several approaches for detecting malaria at an early stage are being proposed, but limitations in this domain still exist (*Majid et al., 2020*). The extraction and selection of features are the most important phases of this process. This research focuses on the suitable selection of features.

The research contribution steps are as follows:

In this article, extensive experimentation is performed individually as well as fusion of PHOG with selected parameters such as 20 bin, 180° angle, two pyramid levels (L), and roi [1; 64; 1; 64] and deep features using pre-trained RasNet-18 and ResNet-50 models for features analysis. Based on experimentation, all extracted features are serially fused and the best features are selected using the generalized normal distribution optimization (GNDO) approach. Finally, the classification is performed based on the selected features using kernels of support vector machine (SVM) and neural network classifiers.

The organization of the articles is as: Related work is described in "Related Work", proposed technique steps are elaborated in "Proposed Methodology", achieved outcomes are discussed in "Simulation Results" and "Conclusion" contains a conclusion.

## RELATED WORK

Several works have been done on the detection and classification of malaria (*Das, Mukherjee & Chakraborty, 2015*; *Abdurahman, Fante & Aliy, 2021*; *Rosado et al., 2016*; *Setyawan et al., 2021*), and some of the most recent findings are discussed in this section (*Arowolo et al., 2021*). The size, shape, and texture features of red blood cells are extracted and fed to a multilayer perceptron network model, achieving an accuracy of 89.80% (*Zamli, 2008*). The extracted features using the dynamic time-warping method and a template-matching classifier achieved a classification accuracy of 73.57% (*Khot & Prasad, 2015*). A model is applied for the detection of malaria parasites by extracting histogram-based texture features (*Chavan & Sutkar, 2014*). A model is introduced that extracts features using the Otsu thresholding method and achieves an accuracy of 91% using the KNN classifier (*Malihi, Ansari-Asl & Behbahani, 2013*). A Bayesian classifier is used to categorize Giemsa-stained peripheral blood samples and achieves a 93.3% accuracy (*Anggraini et al., 2011*). The logistic regression-based classification model is presented based on color space features that provide an accuracy of 88.77% (*Mandal et al., 2010*). A multilayer perceptron classifier is applied that extracts wavelet-based features achieving a 77.19% accuracy (*Yunda, Ramirez & Millán, 2012*). The transfer learning models are applied for malaria classification. This method needs to explore the pre-processing method and optimized features selection method to increase the classification outcomes (*Hemachandran et al., 2023*). The VGG-19 model is fine-tuned to classify the malaria cells. This method also requires to adoption of the best features selection method to improve the

accuracy (*Mariki, Mkoba & Mduma, 2022*). The random forest classifier is applied to classify the malaria cells. This method needs to be analyzed with other classifiers based on informative feature selection methods (*Islam et al., 2022*). Multi-headed attention-based transformer method is designed for classification. This method still needs improvement in the detection and classification of the RBC parasite malaria cells (*Oyewola et al., 2022*). The augmented input images and features are derived through a convolutional neural network for classification. This method provided 94.7% classification accuracy. It needs a low-cost computation model with the highest classification outcomes (*Shewajo & Fante, 2023*).

After a thorough review of existing work on malaria cell classification. To overcome the existing limitations/gaps in this research, a novel method is proposed for the classification of the parasite malaria cells. This method improves the quality of the microscopic malaria images using a bilateral filter. After that hand-crafted and deep features are investigated to classify the malaria cells. Extensive experimentation has been carried out for the extraction, selection, and fusion of hand-crafted and deep features. Furthermore, to authenticate the significance of the proposed method statistical experiment is performed in terms of mean and standard deviations to prove that classification results are consistent.

## PROPOSED METHODOLOGY

Microscopic malaria images are noisy; hence, a bilateral filter is used to smooth the image quality. Then, extract features from the fully connected layer named Fc-1000 of pre-trained models with dimensions of N × 1,000, such as ResNet-50 (*Reddy & Juliet, 2019*) and ResNet-18 (*Guo & Du, 2019*). Furthermore, handcrafted PHOG (*Gour & Khanna, 2020*) features of N × 300 dimensions are extracted from each malaria microscopic image. The extracted deep and handcrafted features are serially blended with the N × 2,300 dimension. GNDO (*Rad et al., 2022*) is employed to select the more informative features N × 498 from N × 2,300 features, which are then supplied to the classifiers. Figure 1 presents the basic steps of the proposed model.

In Fig. 1, a bilateral filter is applied to microscopic malaria images for smoothing. The hand-crafted (PHOG) features with the dimension of N × 300 and deep features having N × 2,000 dimensions are extracted. The derived features are fused serially and provide the dimension of N × 2,300. For prominent features selection GNDO method is used, which is trained on N × 2,300 features based on the pre-trained parameters such as 10 total solutions, 2 C1 and C2 coefficients, and 0.9 weight using 100 training epochs. After training, it is observed that the convergence point is achieved with a fitness value of greater than 0.082 and the model is stable on 100 epochs. At that point, the best features vector N × 498 is obtained out of the N × 2,300 features that are used finally to classify the normal/abnormal cells using neural network and SVM classifiers.

### Image smoothing using bilateral filtering

Bilateral filtering is a nonlinear, edge-preserving noise reduction smooth filter (*Tomasi & Manduchi, 1998*). In this case, pixel intensity is replaced with the average weighted intensity from the neighboring pixels. To preserve the sharpness of edges, this weight could

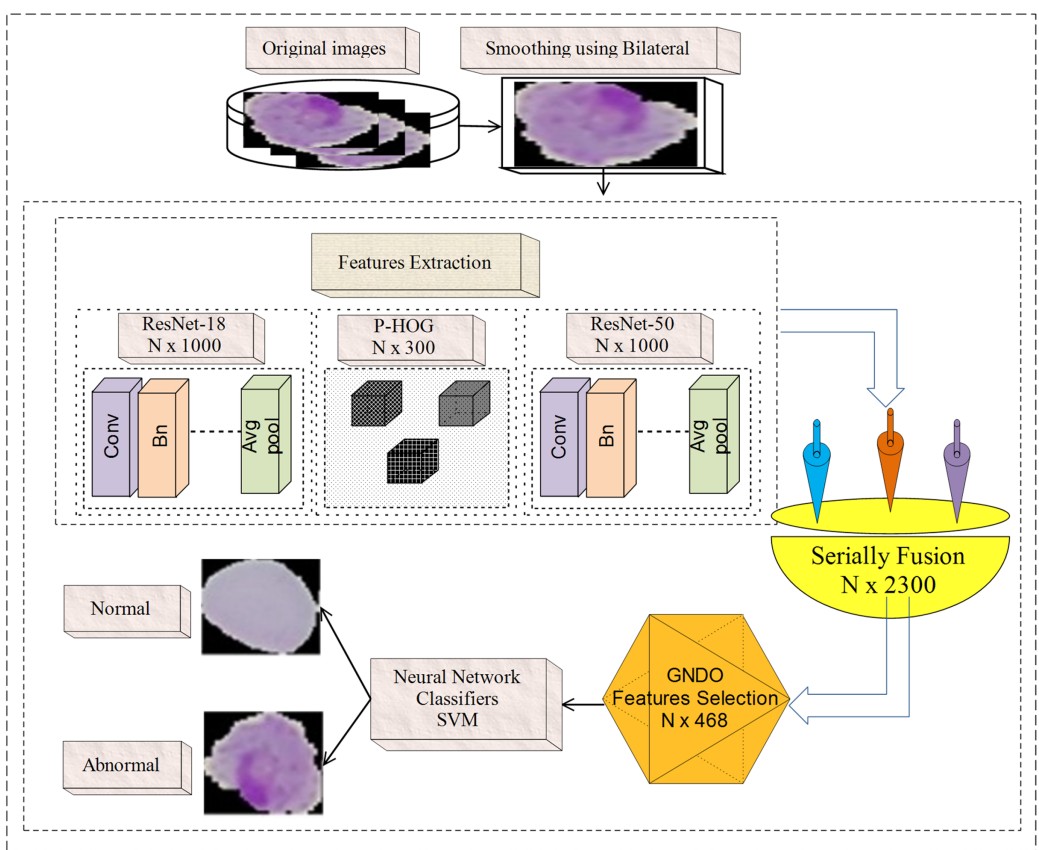

**Figure 1 Malaria classification using handcrafted and deep features fusion.**

rely on Gaussian distribution, Euclidean distance, and radiometric distance (color intensity, distance in depth). The mathematical Eq. (1) of the bilateral filter is:

$$G^{filtered}(x) = \frac{1}{(weight)_p} \sum_{x_g \varepsilon \Omega} G(x_g) f_r(\| G(x_g) - G(x) \|) t_s(\| x_g - x \|), \tag{1}$$

where normalization $(weight)_p$

$$(weight)_p = \sum_{x_g \varepsilon \Omega} f_r(\| G(x_i) - G(x) \|) t_s(\| x_g - x \|).$$

Here, $G^{filtered}$ denotes the filtered output image; $G$, the original microscopic image; $x$, the pixel coordinates; $\Omega$, the cantered point of the window; $f_r$, range of the pixel values to smooth the image intensity; and $t_s$, a spatial kernel to compute the difference of smoothing among the coordinates. Figure 2 presents the smoothing results of the image.

## Deep features extraction

Deep features were derived from smooth images using pre-trained ResNet-18 and ResNet-50 models. The ResNet-18 model consists of 71 layers, which include convolutional (20), ReLU (17), average pool (one), addition (eight), max pooling (one), batch normalization

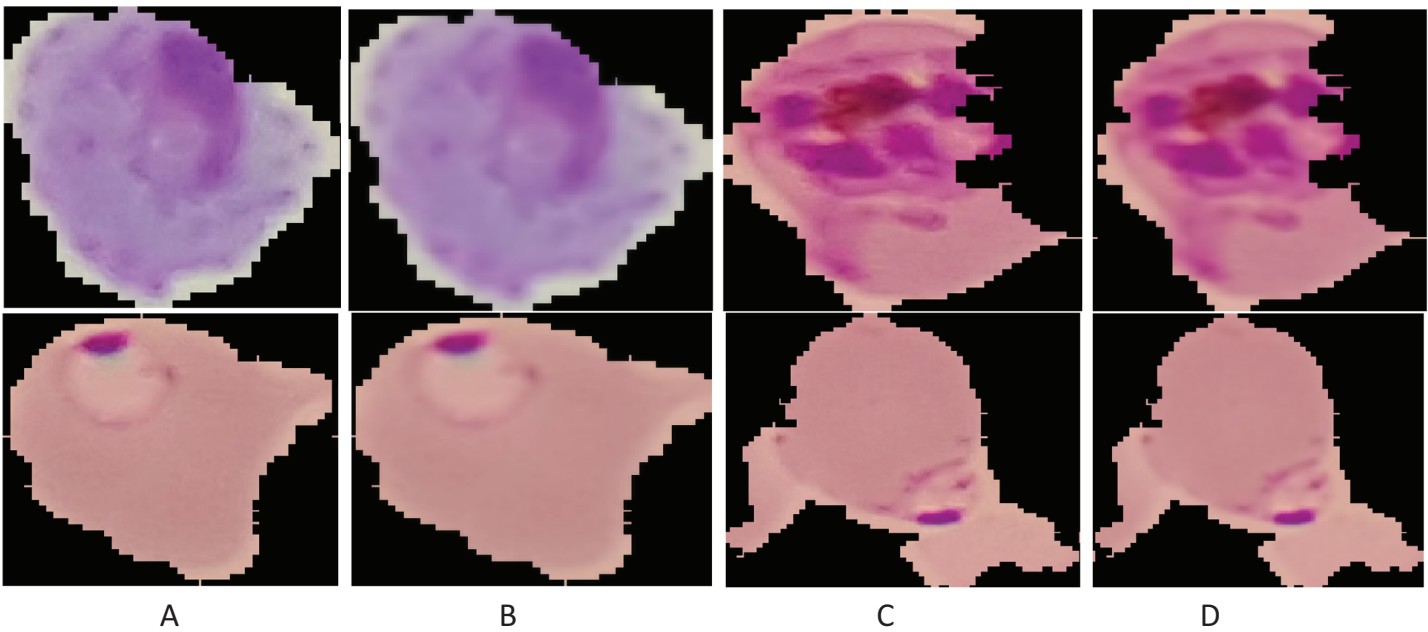

**Figure 2** Enhancement results: (A and C) original images; (B and D) smoothing results.

(20), fully connected, softmax and classification. ResNet-50 model has 177 layers, which include 1 input, (53) convolutional, average pool (one), ReLU (49), batch normalization (53), addition (16), softmax, fully connected, and classification.

The proposed method extracts feature from the fully connected layer with the dimension of $N \times 1,000$ from both pre-trained models and serially fused with $N \times 2,000$ dimensions and supplied to GNDO for the most prominent features selection.

## Feature extraction using PHOG

The shape level and spatial features were extracted by PHOG from each microscopic malaria image with the dimension of $N \times 300$ (*Beulah & Divya Bharathi, 2016*). All images are divided into small grids of pyramids, similar to a quadtree, during this procedure. However, each image exhibits a spatial pyramid-like structure. Every small grid in this structure models a different layer. Figure 3, presents HOGs that are computed across each grid. Then, by merging all these HOGs, we obtain PHOG.

In Fig. 3, PHOG provides help in the reorganization of the lesion shape by computing the orientation of the gradient in a local region.

Table 1 presents the selected parameters of the PHOG features.

HOG features with the dimension of $N \times 300$ are fused serially to the deep feature vector dimension of $N \times 2,000$. The final fused feature vector is $N \times 2,300$ in dimension and is put into the GNDO for the selection of more informative features.

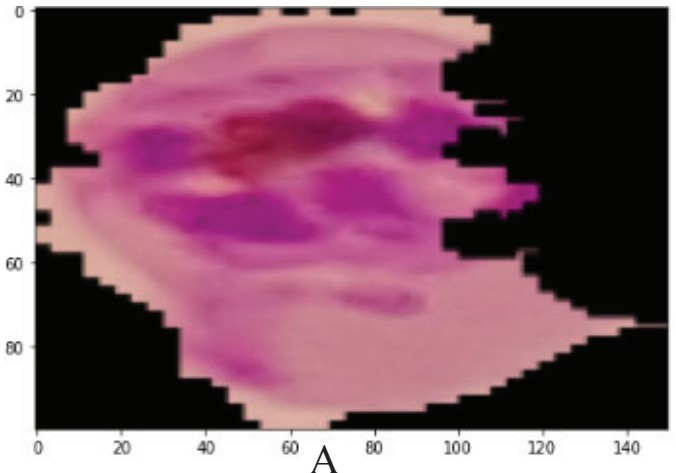 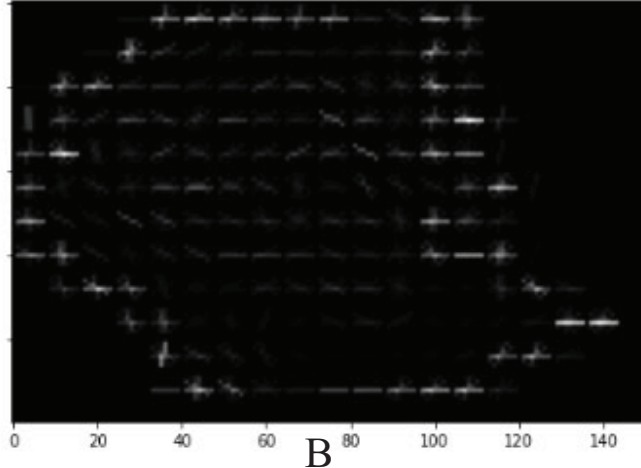

A                 B

**Figure 3 Visualization PHOG results (A) input image (B) PHOG descriptor.**     

**Table 1 The selected parameters of the PHOG features.**

| Parameters | Value |
| --- | --- |
| Bin | 20 |
| Angle | 180 |
| Number of pyramid levels (L) | 2 |
| Region of interest (roi) | [1; 64; 1; 64] |

## Best feature selection using GNDO

In this research, a feature vector dimension of N × 2,300 is supplied to the GNDO, and the optimum feature dimension of N × 468 is selected. GNDO is based on the normal distribution theory. The normal distribution theory is employed to identify and describe significant natural occurrences. Let a variable $x$ follow a probability distribution location $\mu$, and scale parameters are defined in Eq. (2):

$$f(x) = \frac{1}{\sqrt{2\pi}\sigma} exp^{-\frac{(x-\mu)^2}{2\sigma^2}}. \tag{2}$$

Here, $x$ is a normal random variable, and this is referred to as the normal distribution. Equation (2) shows two normal distribution parameters, namely, the location variable $\mu$ and the scaling variables. The location variable $\mu$ denotes the mean value, and the scale variable $\sigma$ denotes the variance. When $\sigma$ remains constant, the probability density curve will shift to either a high or low mean value. However, when the mean value remains constant, the curve will shift toward an increase in the variance. The population-based optimization search process is categorized into three stages. The scattered distribution leads to a global optimum solution, which is then gathered around to achieve the optimum solution. The position of an individual in a normal distribution can be thought of as a

**Table 2 Selected pre-trained parameters for generalized normal distribution optimization model training.**

| | |
|---|---:|
| Total solutions | 10 |
| Total iterations | 100 |
| Coefficient (C1 and C2) | 2 |
| Weights | 0.9 |

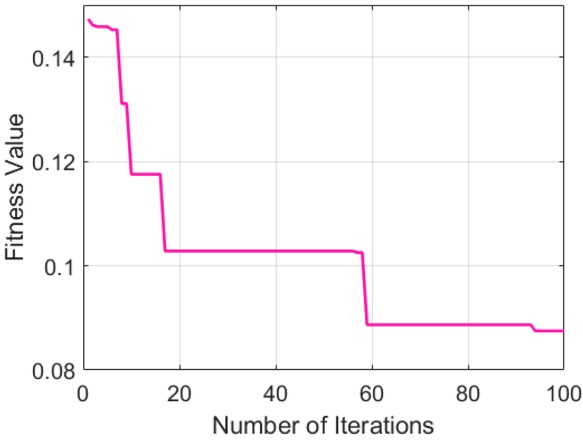

**Figure 4 GNDO model training.**

random variable. The standard deviation of an individual's position in the initial stage is high because the mean value and optimal position value are significantly far off at this point. In the next stage, the mean and the optimal position are much closer to the previous stage. The GNDO model was trained on the pre-trained GNDO parameters with a positive effect on malaria cell classification, as presented in Table 2.

The selected GNDO parameters provide help in determining the optimal fitness values and convergence points. Figure 4 presents a graphical representation of the convergence of the GNDO model over a hundred training epochs.

## Visualization of extracted features using tSNE

tSNE is applied for dimensionality reduction and visualization of high dimensional data. In this article manifold learning method is applied for dimensionality reduction and the complete process is presented in Fig. 5.

In the proposed method, a bilateral filter is used to enhance the image quality. The feature analysis is performed using hand-crafted (PHOG) and deep features (ResNet-18 and ResNet-50). The classification is performed based on the linear SVM using individual features of PHOG, ResNet-18, and ResNet-50 as well as based on fused features of ResNet-18 and ResNet-50 models. Finally, to improve the classification results, hand-crafted (PHOG) and deep features such as ResNet-18 and ResNet-50 are fused serially, and prominent features are selected using GNDO. The selected features are used for further experimentation based on SVM linear, SVM quadratic, SVM cubic, NN-narrow, NN-wide,

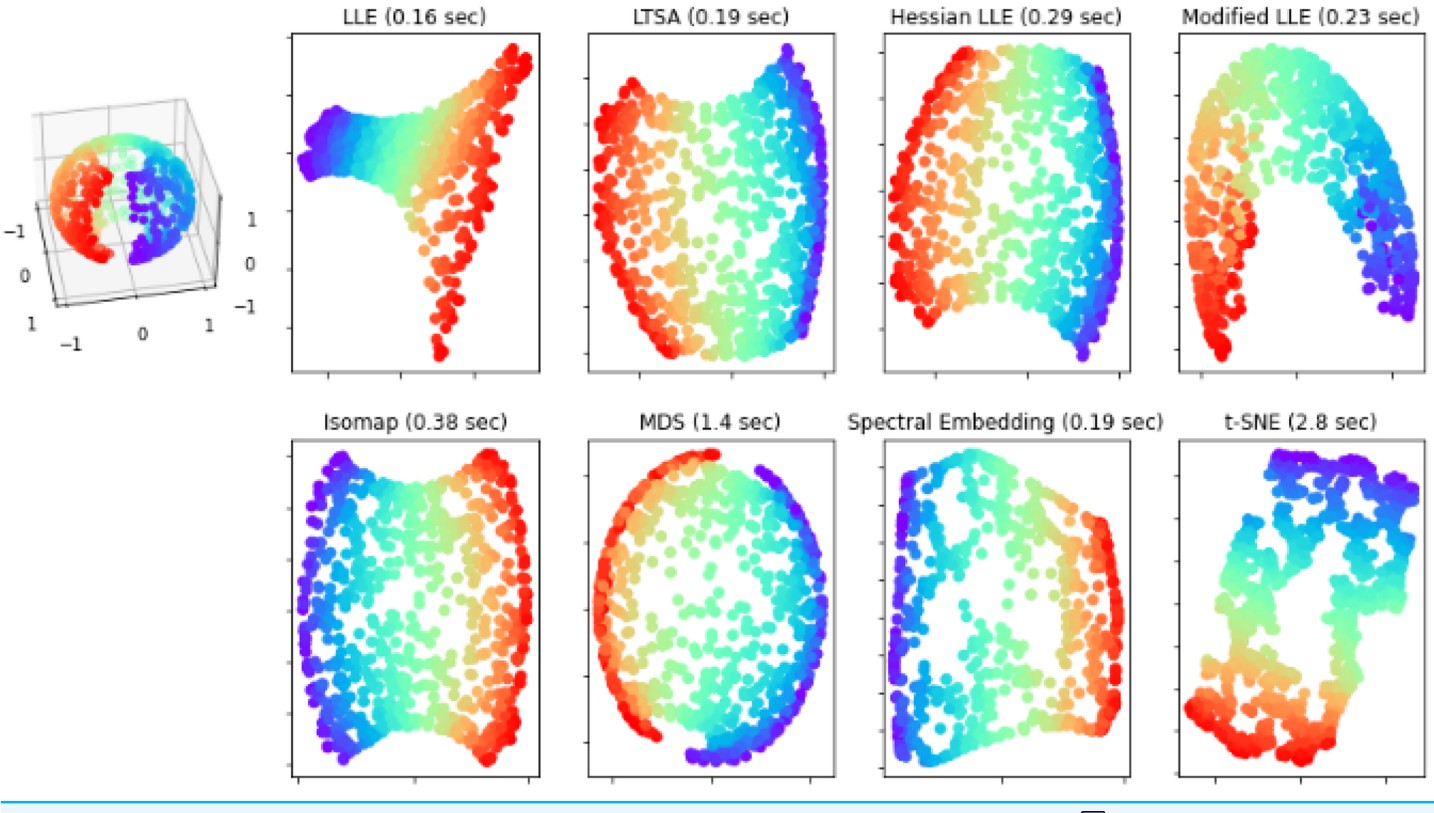

**Figure 5 Shows tSNE of the extracted features.**

| Table 3 Classification results using hand-crafted (manual) (PHOG) and deep features. | |
|---|---|
| **System** | **LAPTOP CORE-I7** |
| Graphic card/graphical processing unit (GPU) | Nvidia 2070 RTX graphic card |
| Software | MATLAB 2021 RA |
| Operating system | Window-11 |
| Generation | 10th generation |

and NN-medium classifiers. To authenticate the classification results, the statistical inference test is performed in terms of mean and standard deviation.

## SIMULATION RESULTS

The classification experiments are conducted based on 10-fold cross-validation to classify infected/uninfected malaria cells. The whole data is divided into ten folds, in which, every iteration is used onefold for the testing set, and other k-1 sets (nine sets) are used for the training set. The RGB microscopic images parasite malaria dataset is publicly available (*Narayanan, Ali & Hardie, 2019*). The dataset contains two classes, infected and uninfected, having each class 13,779 images with a dimension of $256 \times 256 \times 3$. The experimental setup is presented in Table 3.

**Table 4 Proposed method classification results.**

| Extracted features | Features dimension |
| --- | --- |
| ResNet-18 | N × 1,000 |
| ResNet-50 | N × 1,000 |
| PHOG | N × 300 |
| Fused features of (ResNet-18 and 50 models) | N × 2,000 |
| Best features selected using GNDO after fusion of PHOG, ResNet-18, and ResNet-50 features | N × 468 |

**Table 5 Malaria classification results on SVM using 10-fold in terms of ROC.**

| Features extraction method | Classifier | Infected | Uninfected | Accuracy (%) | Sensitivity (%) | Specificity (%) | F1-score (%) |
| --- | --- | --- | --- | --- | --- | --- | --- |
| PHOG | Linear SVM | ✓ | | 0.89 | 0.87 | 0.85 | 0.86 |
| | | | ✓ | 0.88 | 0.85 | 0.87 | 0.85 |
| ResNet18 | | ✓ | | 0.87 | 0.84 | 0.85 | 0.84 |
| | | | ✓ | 0.87 | 0.85 | 0.84 | 0.83 |
| ResNet50 | | ✓ | | 0.82 | 0.81 | 0.80 | 0.80 |
| | | | ✓ | 0.82 | 0.82 | 0.81 | 0.81 |
| Features fusion of ResNet18, and ResNet50 | | ✓ | | 0.90 | 0.87 | 0.88 | 0.89 |
| | | | ✓ | 0.89 | 0.86 | 0.85 | 0.88 |

## Experiment-1: classification of the malaria cells

The classification outcomes are calculated on deep and hand-crafted features separately and fusion of both types of features. The detailed experiment of this research work is mentioned in Table 4.

Table 4 provides the details of features extraction, fusion, and selection along with their dimension. The experiment is performed individually based on ResNet-18, ResNet-50, PHOG, Fusion of ResNet-18, and ResNet-50 features, and selected features using GNDO after fusion of PHOG, ResNet-18, and ResNet-50 features. The outcomes are detailed in Table 5.

In Table 5, malaria cells were classified using manual features and deep features individually based on a linear SVM classifier. Which achieved an accuracy is 0.89 on PHOG, 0.87 on ResNet-18, 0.82 on Res-Net-50, and 0.90 on fusion of the ResNet-18+ Res-Net-50 features.

In the subsequent experiments, the best features are chosen using GNDO by combining features from various sources, including PHOG, ResNet-18, and ResNet-50. The outcomes of these experiments are detailed in Table 6.

### Visual representation of the deep feature learning patterns

Gradient class-weighted activation mapping (Grad-CAM) must explain the capability that could be used to facilitate prediction understanding through the neural network. Grad-CAM is a generalization method that computes the neuron significance in the model

**Table 6 Malaria classification results on 10-fold using NN in terms of ROC.**

| Family | Classifier | Accuracy (%) | Sensitivity (%) | Specificity (%) | F1-score (%) |
|---|---|---|---|---|---|
| Geometric | Linear SVM | 0.95 | 0.97 | 0.95 | 0.96 |
| | | 0.95 | 0.95 | 0.97 | 0.96 |
| | Quadratic SVM | 0.97 | 0.98 | 0.97 | 0.97 |
| | | 0.97 | 0.97 | 0.98 | 0.97 |
| | Cubic SVM | 0.99 | 0.99 | 0.99 | 0.99 |
| | | 0.99 | 0.99 | 0.99 | 0.99 |
| Neural network | Narrow neural network | 0.99 | 0.99 | 0.99 | 0.99 |
| | | 0.99 | 0.99 | 0.99 | 0.99 |
| | Medium neural network | 0.99 | 0.99 | 1.00 | 0.99 |
| | | 0.99 | 1.00 | 0.99 | 0.99 |
| | Wide neural network | 0.99 | 0.99 | 1.00 | 0.99 |
| | | 0.99 | 1.00 | 0.99 | 0.99 |

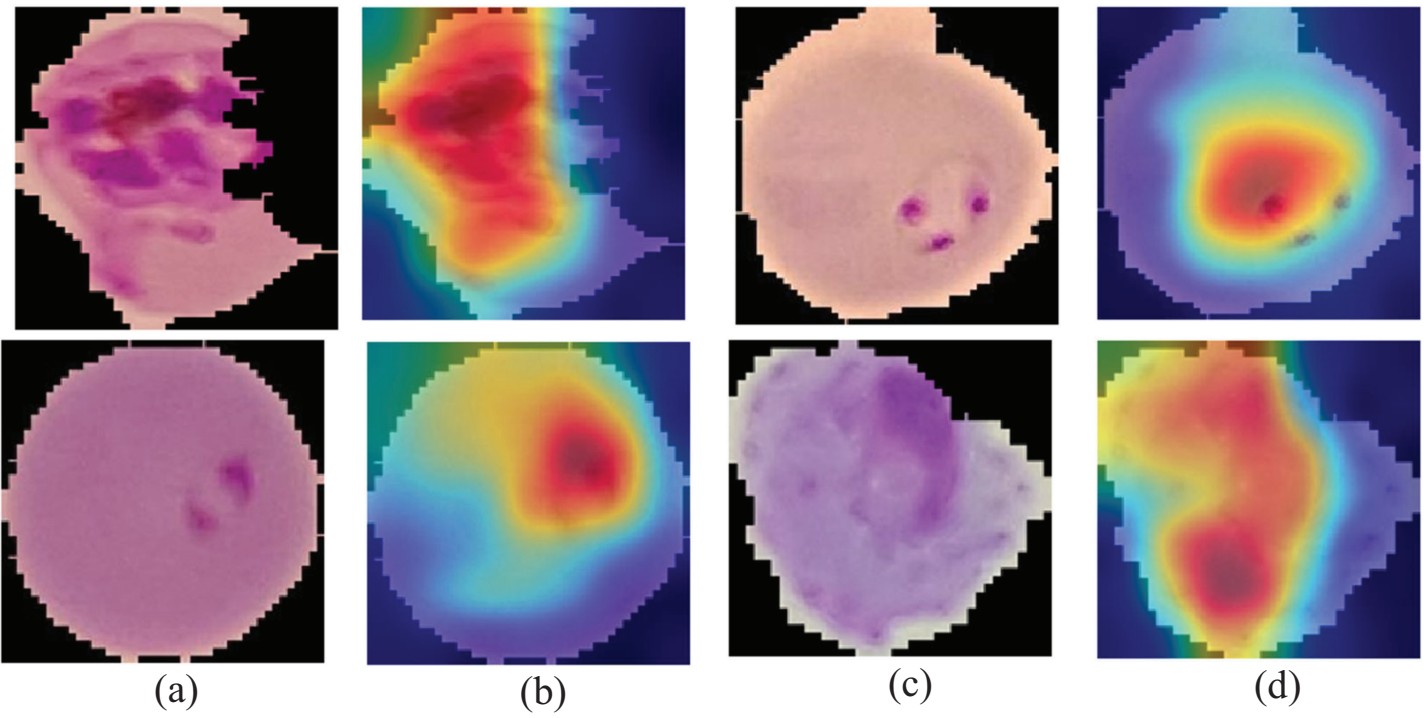

(a)   (b)   (c)   (d)

**Figure 6 Learning patterns of the malaria cells on the average pool using Grad-CAM (A and C) and microscopic images (B and D).**

prediction using gradients as a target. It measures gradient output differently, such as the class score connected to convolutional layer features. The importance of the neuron weights is discovered using a pooled spatial gradient. These weights are used to integrate linear maps of the activation and determine which features played a vital role in the model prediction.

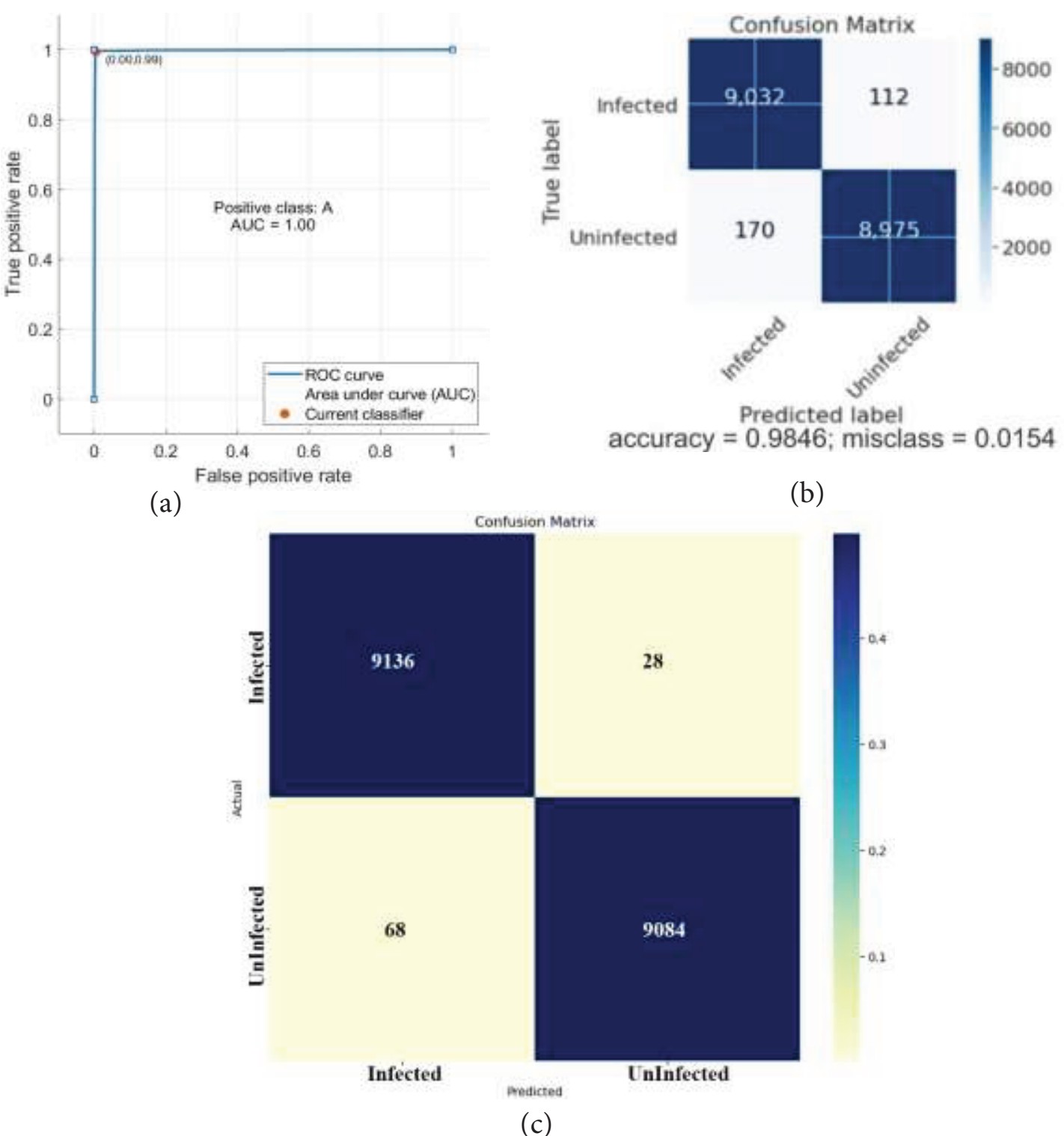

**Figure 7 Classification results.** (A) ROC (B) confusion matrix.

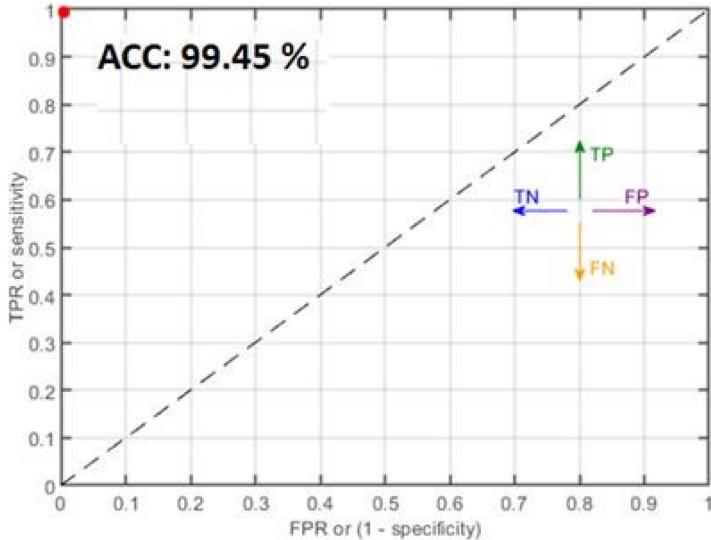

**Figure 8 Classification results where red dots denote the 99.45% accuracy.**

The classification models with $y^c$ output denotes class score c and Grad-CAM (*Selvaraju et al., 2017*) mapping on each convolutional layer with the k number of feature channels, $A_{i,j}{}^k$, in which $i$ and $j$ specify the pixel indices. The weight of a neuron's importance is defined in Eq. (3).

$$\alpha^{channel}{}_k = \frac{1}{N}\sum_i \sum_j \underbrace{\frac{\partial y^c}{\partial A_{i,j}{}^k}}_{Gradient\ by\ backpropagation}.$$

(3)

Here, $N$ denotes the total pixels, and a weighted combination of feature mapping with ReLU is explained in Eq. (4)

$$ReLU = \left(\sum_k \alpha^{channel}{}_k A^k\right).$$

(4)

The activation function ReLU returns the positive features to a given class, and the output is presented in the form of a heatmap of a specified class of the same size as the feature mapping. The Grad-CAM mapping is unsampled to the input size. Figure 6 presents the learning patterns.

The classification outcomes are presented in Fig. 7.

The classification outcomes are calculated on SVM and NN, as demonstrated in Table 6 and Fig. 8.

Table 6 presents the malaria classification outcomes using two family classifiers: geometric and neural networks. On the geometric family, linear, quadratic, and cubic SVM are used to compute classification results with accuracies of 95.71%, 97.38%, and 99.05%, respectively. On the neural network family, narrow, medium, and wide neural network

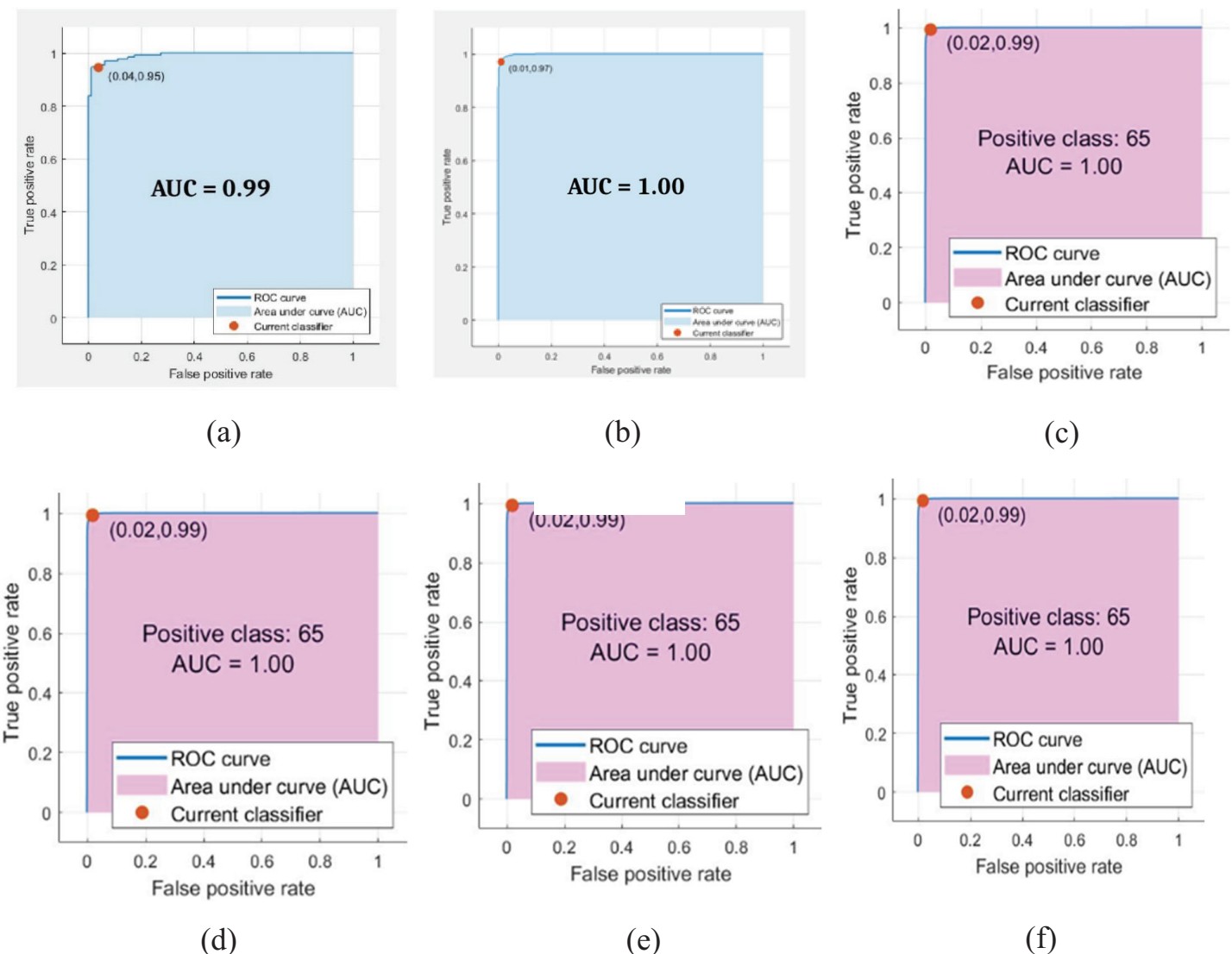

**Figure 9 ROC on benchmark classifiers.** (A) Linear (B) cubic (C) quadratic (D) narrow (E) medium (F) wide.

kernels were used, yielding accuracies of 99.34%, 99.45%, and 99.48, respectively. In this experimental analysis, we observed that the neural network's wide kernel provides maximum accuracy.

## Statistical inference test for malaria classification

The proposed classification method is statistically analyzed on selected kernels, such as (linear, quadratic, cubic), (narrow, medium, and wide) of SVM and neural network classifiers, as presented in Figs. 10 and 11.

Tables 7 and 8 present the quantitative classification results.

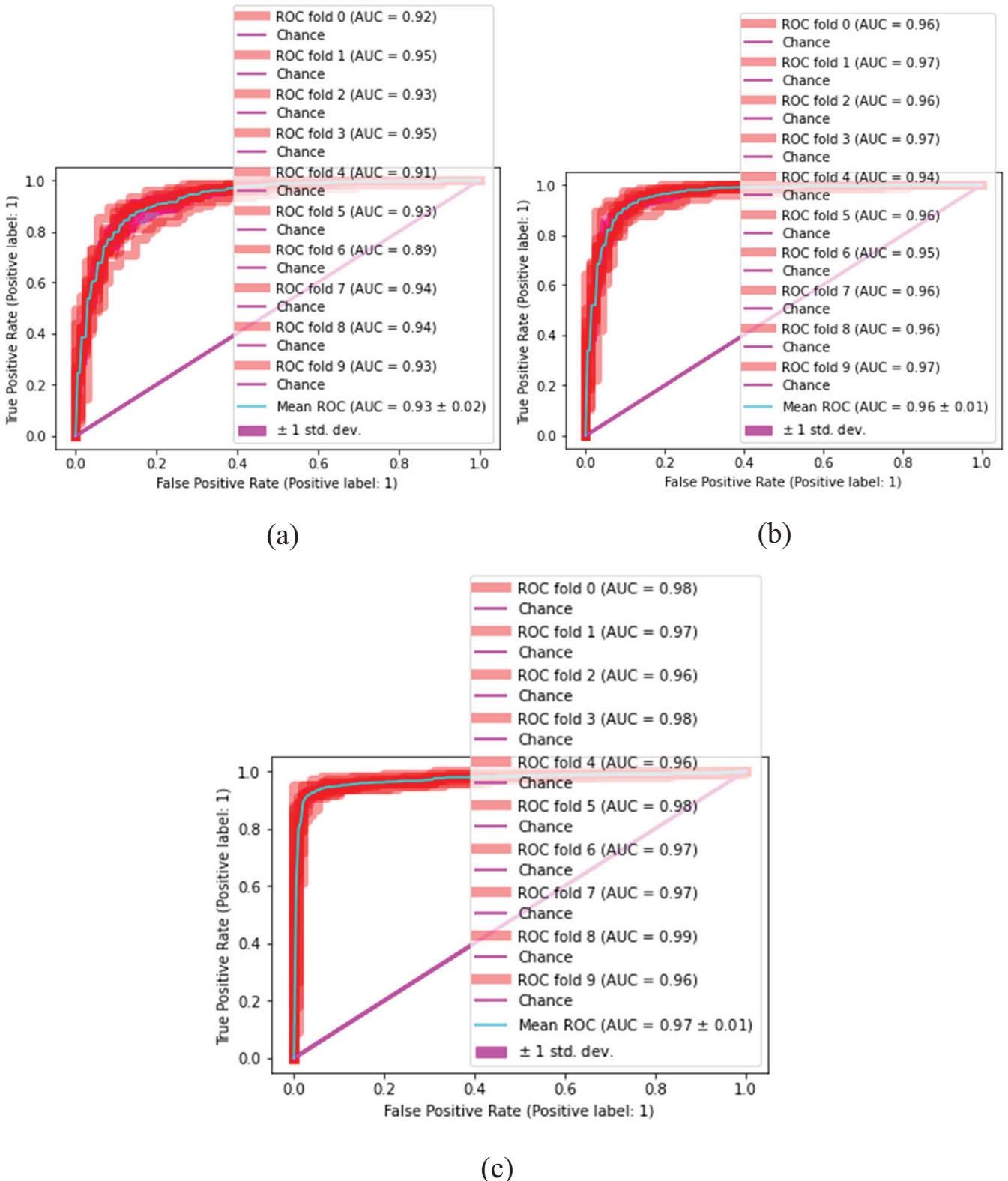

**Figure 10 Malaria classification results.** (A) SVM-linear, (B) SVM-cubic, (C) SVM-quadratic.

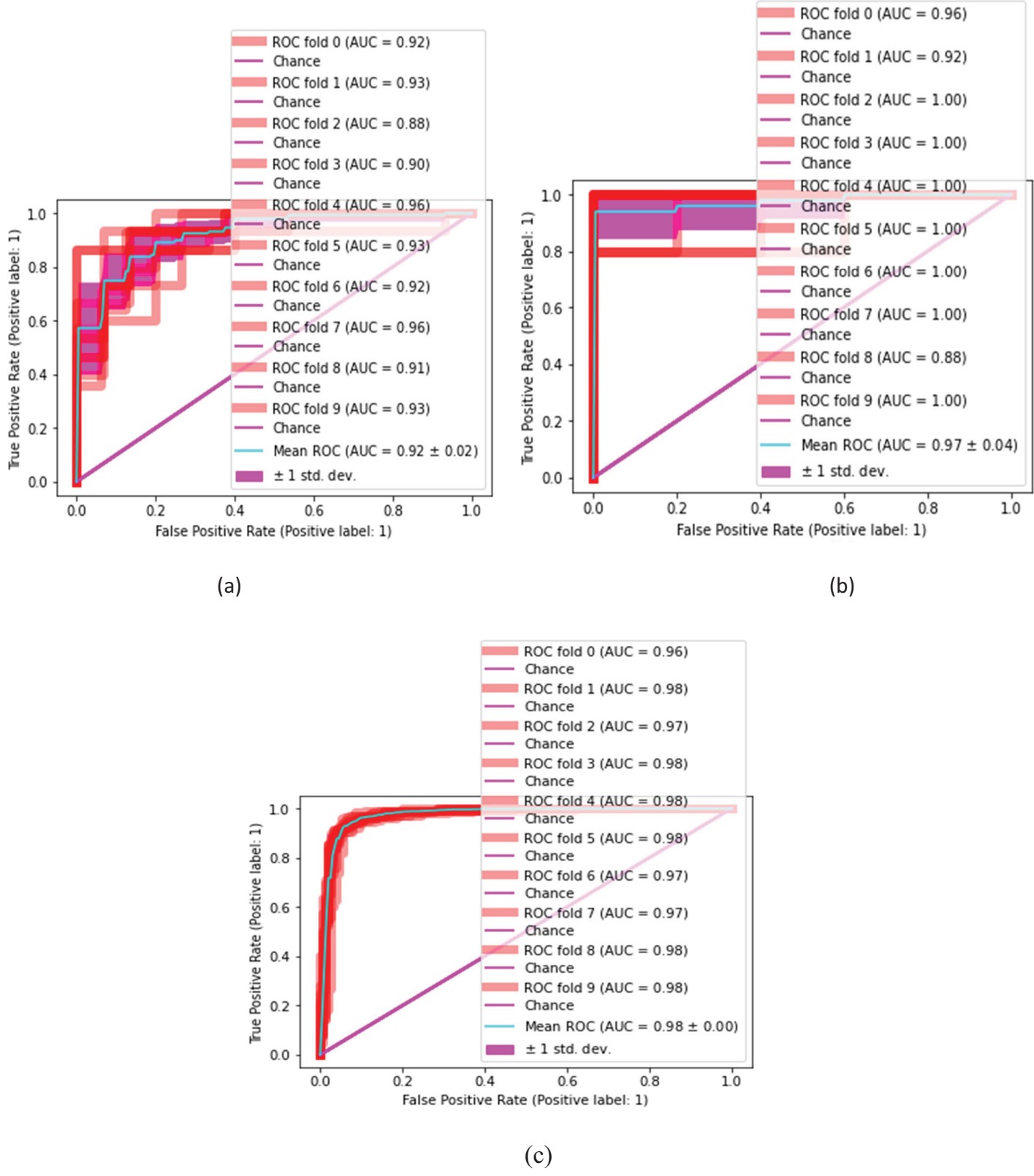

**Figure 11 Malaria classification results by NN.** (A) NN-narrow, (B) NN-medium, (C) NN-wide.

**Table 7 Comparison of testing results in terms of ROC.**

| | | | | | | | | | | | |
|---|---|---|---|---|---|---|---|---|---|---|---|
| SVM-Linear | 0.92 | 0.95 | 0.93 | 0.95 | 0.91 | 0.93 | 0.89 | 0.94 | 0.94 | 0.93 | 0.93 ± 0.02 |
| SVM-quadratic | 0.96 | 0.97 | 0.96 | 0.97 | 0.94 | 0.96 | 0.95 | 0.96 | 0.96 | 0.97 | 0.96 ± 0.01 |
| SVM-cubic | 0.98 | 0.97 | 0.96 | 0.98 | 0.96 | 0.98 | 0.97 | 0.97 | 0.99 | 0.96 | 0.97 ± 0.01 |

**Table 8 Proposed result comparison using infection malaria dataset.**

| | | | | | | | | | | | |
|---|---|---|---|---|---|---|---|---|---|---|---|
| NN-narrow | 0.92 | 0.93 | 0.88 | 0.90 | 0.96 | 0.93 | 0.92 | 0.96 | 0.91 | 0.93 | 0.92 ± 0.02 |
| NN-medium | 0.96 | 0.92 | 1.00 | 1.00 | 1.00 | 1.00 | 1.00 | 1.00 | 0.88 | 1.00 | 0.97 ± 0.04 |
| NN-wide | 0.96 | 0.98 | 0.97 | 0.98 | 0.98 | 0.98 | 0.97 | 0.97 | 0.98 | 0.98 | 0.98 ± 0.00 |

**Table 9 Proposed result comparison using infection malaria dataset.**

| Ref# | Year | Results % |
|---|---|---|
| Vijayalakshmi (2020) | 2020 | Accuracy = 93.13, Sensitivity = 93.44 |
| Maqsood et al. (2021) | 2021 | Accuracy = 91.00, Sensitivity = 91.23 |
| Elangovan & Nath (2021) | 2021 | Accuracy = 98.35, Sensitivity = 97.67 |
| Sadafi et al. (2021) | 2021 | Accuracy = 88.18, Sensitivity = 87.19 |
| Tavakoli et al. (2021) | 2021 | Accuracy = 94.65, Sensitivity = 92.21 |
| Pimple, Likhitkar & Pande (2022) | 2022 | Accuracy = 95.91, Sensitivity = 94.52 |
| Bhuiyan & Islam (2023) | 2023 | Accuracy = 97.92 |
| Almurayziq et al. (2023) | 2023 | Accuracy = 91.60 |
| Alqudah, Alqudah & Qazan (2020) | 2020 | Accuracy = 98.90, Sensitivity = 98.81 |
| **Proposed method** | | **Accuracy = 99.00, Sensitivity = 100.00** |

In this experiment, the cubic kernel of the SVM classifier produces a mean ROC of 0.97 ± 0.01, whereas the linear and quadratic kernels of SVM produce mean ROCs of 0.93 ± 0.02 and 0.96 ± 0.01, respectively. Figure 9 presents the malaria classification results using NN. Table 8 presents the quantitative evaluation of NN.

In Table 8, the wide kernel of NN provides a maximum mean ROC of 0.98 ± 0.00 on the neural network classifier. Table 9 presents the proposed testing results in terms of ROC using different kernels of NN and SVM classifiers.

Table 9, depicts classification results that are computed on different kernels of NN and SVM classifiers which we achieved using NN 0.98 ± 0.00 ROC on wide kernel and 0.97 ± 0.01 ROC on SVM quadratic. Table 10 provides a detailed comparison of classification outcomes with the existing methods.

In Table 10, five recent works are considered for comparison with the proposed method. Traditional techniques are employed to obtain the classical features, which are then passed to the SVM classifier. This work provided an accuracy of 93.13% (Vijayalakshmi, 2020). The features are derived from classical handcrafted techniques and achieved

**Table 10 Malaria classification results of proposed method in terms of ROC using different kernels of NN and SVM classifiers based on 10 fold cross-validation.**

| Kernel of NN | | | Kernel of SVM | | |
|---|---|---|---|---|---|
| Narrow | Medium | Wide | Linear | Cubic | Quadratic |
| 0.92 ± 0.02 | | | | | |
| | 0.97 ± 0.04 | | | | |
| | | 0.98 ± 0.00 | | | |
| | | | 0.93 ± 0.02 | | |
| | | | | 0.96 ± 0.01 | |
| | | | | | 0.97 ± 0.01 |

a 91% accuracy by using a naïve Bayes classifier (*Maqsood et al., 2021*). In this work, the features were extracted using different handcrafted methods and achieved an accuracy of 98.35% (*Elangovan & Nath, 2021*). The fusion of CNN and handcrafted features gives an accuracy of 88% (*Sadafi et al., 2021*). The features in this work are derived from the handcrafted method and then classified with 92% accuracy using the SVM classifier.

This research provides an improved classification model, as compared with existing work, by fusing derived deep and handcrafted features using the ResNet18 model, ResNet50 pre-trained model, and PHOG handcrafted feature extraction method and then selecting the most prominent features using GNDO for the classification of malaria cells.

## CONCLUSION

Precise and intelligent detection of malaria parasitic cells at early stages is a challenging task. To address the challenges/limitations, a bilateral filter is applied to improve the quality of microscopic malaria images. From the enhanced images, PHOG features are extracted based on selected parameters such as 20 bins, 180° angle, two pyramid levels (L), and roi [1; 64; 1; 64] as well as deep features are also extracted from ResNet-18 and ResNet-50 models. All the extracted features are serially fused. The best features are selected using GNDO having the dimension of N × 498 features and used for classification based on SVM and neural network classifiers. Best results are achieved such as 99.48% accuracy, 0.99 sensitivity, 1.00 specificity, and 0.99 F1-score using a wide neural network classifier because of the designed architecture of the proposed method.

### Limitations and future work

This work is limited to binary classification of the parasite malaria microscopic images. In the future, this research may be extended for multiclassification of parasite malaria using real patient data from several acquisition sources. Nowadays quantum computing gaining more importance as compared to machine/deep learning. Therefore, this work may also be extended using quantum machine learning for the multi-classification of parasite malaria cells.

### Funding

This research was supported by the National Research Foundation of Korea (NRF) grant funded by the Korea government (MSIT) (No. 2021R1A4A1031509). The APC was funded by the Technology Development Program of MSS (No. S3033853). The funders had no role in study design, data collection and analysis, decision to publish, or preparation of the manuscript.

### Grant Disclosures

The following grant information was disclosed by the authors:
National Research Foundation of Korea (NRF): 2021R1A4A1031509.
Technology Development Program of MSS: S3033853.

### Competing Interests

The authors declare that they have no competing interests.

### Author Contributions

- Javeria Amin conceived and designed the experiments, analyzed the data, prepared figures and/or tables, and approved the final draft.
- Muhammad Almas Anjum analyzed the data, performed the computation work, authored or reviewed drafts of the article, and approved the final draft.
- Abraz Ahmad conceived and designed the experiments, performed the computation work, prepared figures and/or tables, and approved the final draft.
- Muhammad Irfan Sharif conceived and designed the experiments, authored or reviewed drafts of the article, and approved the final draft.
- Seifedine Kadry performed the experiments, analyzed the data, prepared figures and/or tables, and approved the final draft.
- Jungeun Kim performed the computation work, authored or reviewed drafts of the article, and approved the final draft.

### Data Availability

    The data are available at the National Library of Medicine: https://lhncbc.nlm.nih.gov/LHC-downloads/downloads.html#malaria-datasets.

### Supplemental Information

Supplemental information for this article can be found online at http://dx.doi.org/10.7717/peerj-cs.1744#supplemental-information.

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
