# Peer review of "Microscopic parasite malaria classification using best feature selection based on generalized normal distribution optimization"

_PeerJ Computer Science, doi:10.7717/peerj-cs.1744_

## Round 0.1 · original submission · Major Revisions

Dear authors,

Your article has not been recommended for publication in its current form. However, we do encourage you to address the concerns and criticisms of the reviewers and resubmit your article once you have updated it accordingly.

Best wishes,


Reviewer 1 ·

Basic reporting

I have reviewed your work titled “Microscopic parasite malaria classification using best feature selection based on generalized normal distribution optimization” in detail. I would like to state that the proposed method in the study is innovative. However, I saw that there were major deficiencies in the study.
In their study, the researchers performed feature extraction using Resnet18, Resnet50, and PHOG methods. They extracted 1000 features each from Resnet18 and Resnet50 layers and 300 features by the PHOG method. Later, the number of combined features was 2300. Then, 498 feature selections were made with the GNDO method. However, before these processes were performed, the image enhancement step was applied to the data set and this is not specified in the abstract. The abstract should be written more fluently. In Figure 1 and line 117, it is understood that feature extraction is done from the avaregepooling layer. But I think feature extraction is done from Fullyconnected layer. This error needs to be corrected in the relevant places. It is important to write the proposed model in detail. The flow chart of the proposed model and the GNDO method should be presented in detail.

Experimental design

Information should also be given about the source from which the data set is taken, how many classes, and how many images it consists of. I don't understand the logic of Table 7. Confusion matrices of the highest results obtained in the study can be added to the study. Limitations of the study should be presented. Finally, the number of data in the study starts with N in the abstract and ends with N in the results.

Validity of the findings

How many images were used in the training process and how many images were used in the testing process? Also, for example, what would be the result if the PHOG method was not used? What is the effect of the PHOG method on the proposed model? It is important to present the results obtained in the PHOG, Resnet18, and Resnet50 methods in order to eliminate the deficiencies in the model part. The step-by-step progress of the study will avoid confusion. It is important to eliminate these deficiencies in the study.

Additional comments

Researchers should at least compare the model they propose with the pre-trained models they use as a base in the study. Examples are Resnet18, Resnet50, etc. Also, what effect does the data preprocessing step really have on the results, would the results change without this step?

·

Basic reporting

Line 74 - Problem Statement (clear): The researcher was able to identify other researchers who have done work on a similar area, however, there are still problems since malaria input images comprise noise, poor contrast, lighting difficulties, and stain. The author noted that informative feature selection for malaria classification remains a difficult task due to low-quality images, quick variations, size, and irregularity within the region of interest. The researcher went further to say that all of these have been addressed in this proposed work.

Note: Clearly show how you addressed all these problems in your work.


Line 83 – Aim (well stated): This research focuses on the suitable selection of features (the is in -line with the paper title). The extraction and selection of features is the most critical phase of this process of malaria detection as stated by the author.

Line 84 - The research contribution (clearly shown): the researchers employed the extraction of handcrafted features, using PHOG, and deep features using the pre-trained Rasnet-18 and ResNet-50 models to accomplish the analysis of the comprehensive features. Furthermore, serially fused these features out of which selected optimum features from the pool of extracted features using the generalized normal distribution optimization (GNDO) approach. Line 88 - The researcher classified malaria cells using kernels of SVM such as [linear, quadratic, cubic] and [narrow, medium, and wide] of the neural network classifier.

Line 305 - Result: The authors showed that the microscopic malaria cells were improved using a bilateral filter. Deep features from ResNet-50 and ResNet-18 models and handcrafted PHOG features were extracted from each malaria cell and serially merged.

Line 306 shows that GNDO is used to select the best N × 498 features out of N × 2,300 and input them into the SVM/neural network classifiers in which a wide neural network provides 99.48% accuracy, 0.99 sensitivity, 1.00 specificity, and
308 0.99 F1-score.


I wish to raise the following concerns, which I think that when corrected will help to improve the work.

Line 22: correct the topographical error on the highlighted part. i.e parasites instead of sparasites.
Don’t use acronyms that you didn’t define
Lines 23 & 24, 66, 69 & 70: Don’t use acronyms before defining it, e.g. PHOG - RESNET=50, -RESNET -18, PCA, KNN, SVM, and DL
Line 23: Add ’s’ in ‘method’ at
L45: put a ‘full stop’ after transfusion
Use numbering styles of citation to conserve space, e.g. [1,2]
You may recast the sentence in 70 to enhance comprehension.
Line 74 may read the malaria input image.
Line 79: source? Which region of the world? Support this assertion with reliable literature. Note: malaria is not equally distributed in all regions of the world. Be specific, you don’t need to generalize this.
Line 84: add ‘follows’ after as:
Line 96 (citation style): Be consistent in your citation style e.g. you used et al in some cases while on the other hand, you listed as many as five authors for one citation, this litters your work and consumes a lot of space.
Line 97, 98: The cited paper in highlighted portion look alike, if so, remove one.
Line 210: Does the dataset (13779) images cited belong to the researcher cited? If it is your dataset, recast the sentence and cite it appropriately.
Information about the dataset used was not discussed in your methodology ie data source, method of collection, and data size. your methodology needs more details.

Conclusion: The researchers were able to show how malaria cells were classified using manual features and deep features individually based on a linear SVM classifier as mentioned in Table 3.

Experimental design

The research looks original and the research problem is well formulated.

Validity of the findings

Note: I was not able to access the data file. I couldn't open the data file. I only saw sample malarial blood films available in the paper.

---

## Round 0.2 · Major Revisions

Dear authors,

Thank you for the revised paper. After carefully checking the paper, it seems that you have not responded all of the reviewers’ concerns, questions, and suggestions. Your article has not been recommended for publication in its current form. However, we do encourage you to address the concerns and criticisms that are listed below and resubmit your article once you have updated it accordingly.

1. Before feature selection processes, the image enhancement step was applied to the data set and this is not specified in the abstract. The abstract should be written more fluently.

2. In Figure 1 and line 117, Although feature extraction is done from Fullyconnected layer, it is understood that feature extraction is performed from the avaregepooling layer. This confusion needs to be corrected in the relevant places.

3. It is important to write the proposed model in detail. The flow chart of the proposed model and the GNDO method should be presented in detail.

4. Information should also be given about the source from which the data set is taken, how many classes, and how many images it consists of. The logic of Table 7 is confusing. Confusion matrices of the highest results obtained in the study can be added to the study.

5. Limitations of the study should be presented.

6. It is not clear how many images were used in the training process and how many images were used in the testing process. Also, for example, what would be the result if the PHOG method was not used? What is the effect of the PHOG method on the proposed model? It is important to present the results obtained in the PHOG, Resnet18, and Resnet50 methods in order to eliminate the deficiencies in the model part. The step-by-step progress of the study will avoid confusion. It is important to eliminate these deficiencies in the study.

7. Researchers should at least compare the model they propose with the pre-trained models they use as a base in the study. Examples are Resnet18, Resnet50, etc. Also, what effect does the data preprocessing step really have on the results, would the results change without this step?


Best wishes,

---

## Round 0.3 · Minor Revisions

Dear authors,

Thank you for the revision. The paper seems to address many of the concerns and questions of the reviewers. You urged to respond to the following comments in order to increase the scientific breadth of the paper. You also need to respond to the remarks below in order to raise the presentational quality of your work.


1. Keywords should be written in alphabetical order.

2. Please, provide a paragraph with three to five clear positive impacts of the proposed method.

3. You just described the related works that the researchers have done, but you did not evaluate the advantages and disadvantages of the related works. Regarding the Introduction, what makes this article different from the rest of the studies that are available in the literature is not specified. The gap in existing literature, by arguing what is missing or inadequate in existing solutions and thus your study is necessary is not identified. This needs to be briefly noted and then further elaborated, with in-depth analysis and substantiation of citations.

4. The contributions of this work are not clear. I went through the abstract and introduction, however did not get the main contribution of this work. I suggest you spend significant efforts to enhance the main work in this research. The paragraphs on related works just show the text of the related works without any effect, discussion or overview to give a summary for that text.

5. There are English grammar and writing style errors. The paper also needs proofreading.

6. Clarifying the study’s limitations allows the readers to better understand under which conditions the results should be interpreted. A clear description of limitations of a study also shows that the researcher has a holistic understanding of his/her study. However, the authors fail to demonstrate this in their paper. The authors should clarify the pros and cons of the methods. What are the limitation(s) methodology(ies) adopted in this work? Please indicate practical advantages, and discuss research limitations.

7. Add further details on how simulations were conducted. Similarly, system and resource characteristics could be added to Tables for clarity. The paper lacks the running environment, including software and hardware. The analysis and configurations of experiments should be presented in detail for reproducibility. It is convenient for other researchers to redo your experiments and this makes your work easy acceptance. A table with parameter setting for experimental results and analysis should be included in order to clearly describe them.

8. Important data are missing in the experimental results section so that the experiments can be reproduced, and even so that conclusions can be drawn from the reported results. Parameters of the method are not presented.

9. All variables should be written in italics as in equations. Some equations should be used with correct citations. They seem as if they are proposed and used first in this paper. This approach cannot be acceptable. Furthermore, do not use “as follows”, “as” for equations. Use correct equation numbers instead. The equations are part of the sentences. The punctuation should be corrected.

10. Some mathematical notations are not rigorous enough to correctly understand the contents of the paper. The authors are requested to recheck all the definitions of variables and further clarify these equations. Definitions of all variables and their intervals should be given.

---

## Round 0.4 · Minor Revisions

Dear Authors,

Thank you for your hard work on the revision. However, the paper does not seem to be proofread. There are still errors in English grammar and writing style. The paper definitely needs proofreading. For example see lines: 102-108.

It also appears that some references have been added and misspelled. It is not clear why you have added the references listed below after the last revision, although there are no suggestions, criticisms and concerns. It also appears that one reference has been removed from the list. It is clear that you have added some tables, but you should indicate this situation in the cover letter and rebuttal file.

Best wishes,

- Hemachandran, K.; Alasiry, A.; Marzougui, M.; Ganie, S.M.; Pise, A.A.; Alouane, M.T.-H.; Chola, C. Performance Analysis of Deep Learning Algorithms in Diagnosis of Malaria Disease. Diagnostics 2023, 13, 534. https://doi.org/10.3390/diagnostics13030534
- Islam, M.R.; Nahiduzzaman, M.; Goni, M.O.F.; Sayeed, A.; Anower, M.S.; Ahsan, M.; Haider, J. Explainable Transformer-Based Deep Learning Model for the Detection of Malaria Parasites from Blood Cell Images. Sensors 2022, 22, 4358. https://doi.org/10.3390/s22124358

---

## Round 0.5 · accepted · Accept

Dear authors,

Thank you for addressing all the reviewers' and editor's comments clearly. Your article is now acceptable for publication after the final revision.

Best wishes,